# Monitoring Urban Expansion and Urban Green Spaces Change in Addis Ababa: Directional and Zonal Analysis Integrated with Landscape Expansion Index

**Eyasu Markos Woldesemayat** [1,*] and **Paolo Vincenzo Genovese** [2]

1   School of Architecture, Tianjin University, Tianjin 300072, China
2   Bionic Architecture and Planning Research Centre, School of Architecture, Tianjin University, Tianjin 300072, China; pavic_genovese@tju.edu.cn
*   Correspondence: eyasumark@gmail.com; Tel.: +86-1662-2101-443 or +251-912-63-00-22

**Abstract:** Addis Ababa, the capital of Ethiopia, is urbanizing very fast. This study aimed to assess urban expansion and Urban Green Spaces (UGS) change in the city from 1989 to 2019. Remote Sensing and Geographical Information System (GIS) and Landscape Expansion Index (LEI) were used to extract Land Use Land Cover (LULC) data, measure urban expansion and UGS change and analyze urban growth pattern in inner zone, outer zone and eight quadrants. The results showed that urban area in the inner zone increased from 3712 ha to 3716 ha (0.1%), and from 3716 ha to 3874 ha (4.2%) and in the first (1989–1999) and second periods (1999–2009), while it decreased from 3874 ha to 3733 ha (3.6%) in the third period (2009–2019), portraying a non-unidirectional trend of change. Conversely, the UGS in the inner zone decreased from 60 ha to 54 ha (10%), and from 54 ha to 38 ha (29.6%) in the first and second periods, while it increased from 38 ha to 53 ha (39.4%) in the third period, reporting spatial tradeoff between the two land cover types. Meanwhile, urban areas in the outer zone increased from 10,729 ha to 15,112 ha (40%), from 15,112 ha to 21,377 ha (41.4%) and from 21,377 ha to 28,176 ha (31.8%) in the first, second and third periods, respectively, representing frontiers of suburbanization. On the other hand, the UGS in the outer zone decreased from 3624 ha to 3171 ha, from 3127 ha to 2555 ha and from 2555 ha to 1879 ha, with an annual rate of decline of 1.25%, 1.8% and 2.6%, respectively, showing increasing trend of UGS destruction for urban construction. Furthermore, the LEI analysis result showed that urban expansion pattern demonstrated largely an outlying growth characterized by differentiation and isolation of patches, whereas the infill and edge expansion pattern were insignificant and fluctuated over 30 years. Furthermore, the directional analysis showed that urban area predominately expanded in SEE,> SSE,> SSW,> SWW,> and NEE directions with varying magnitude in the first, second and third period, but decreased in third period in NWW, < NNW< and NNE directions. In response to such urban growth pattern, the center of gravity of urban area shifted from north to south during the study period, displaying main direction urbanization in recent years. Conclusively, zonal and directional studies are more effective in characterizing the Spatio-temporal dynamics variabilities of urban expansion and UGS change for informed urban planning towards sustainable urban development.

**Keywords:** capital city; landscape expansion index; spatio-temporal dynamics; spatial transformation; suburbanization

## 1. Introduction

Urbanization, which is commonly described as social and political changes that result from economic development and industrialization [1], is occurring at an unprecedented rate across the world. According to the United Nations [2], 55% of the total the human population currently lives in an urban area and this figure is projected to surge to 70% by 2050 [3]. Of this growth, 90% of the urban population increase will take place in Africa and Asia, the two largest continents on our planet [4,5]. Unlike Asia, Africa's late but

rapid is characterized by intensive concentration of population, assets and functions [5]. The accelerated urbanization has been consuming natural resources at alarming rate and subsequently has led to many intertwined social, economic, and environmental challenges, i.e., shortage of housing, lack of employment, congestion, lack of basic amenities and mounting environmental challenges. Among environmental challenges, loss of ecological resources, decline of the proportion of the UGS, increases in the urban thermal environment, increases in impervious surfaces and high building density [6], Urban Heat Island (UHI) effects and ecological climate change extremes have been reported in recent studies [7,8]. To make matter worse, urbanization in Africa is accompanied by large-scale informal settlements expansion. Informal settlements expansion in Africa, especially in sub-Saharan countries outweighs planned expansion and accounts for 60–70% of settlement and it is expected to grow exponentially [9]. Unfortunately, most of these expansions have been observed in ecologically sensitive areas [3,4,10,11], such as protected forests, waterways, riverbanks, hilly areas and other green areas. This has contributed to deterioration and decline of UGS due to pressure emanated from the intensive urbanization, as indicated above [11,12].

Like many other African country cities, Addis Ababa, the capital of Ethiopia, is urbanizing very fast due to population increase, changes in economic policy measures, and introduction of urban development policy. The city's urban landscape have been rapidly transformed due to these policy formulation and subsequent implementation over the past decades. Illustratively, the city exhibited two types of urban landscape transformation i.e., out-ward expansion and inner city redevelopment program. The former contributed to rapid horizontal expansion while the later resulted in revitalization of inner neighborhoods, which caused relocation of residents to the public buildings called condominium houses in the peripheral sites. Like many other African country cities, the outward expansion is characterized by haphazard urban growth, which has flushed out UGS available in the city [13,14]. Evidently, spatio-temporal dynamics studies have revealed that the city is experiencing fast decline of the UGS and as a result the city is facing environmental challenges [13–17], namely; shrinking of the urban forest, decline of surface water quality, destruction of cultivable land, rising urban temperature, poor collection and management of solid waste, and a general deterioration of urban environment have been reported in many recent literatures [17,18]. Although, such studies provided evidence on the environmental challenges and shrinking trend of the UGS in Addis Ababa, they failed to identify urbanization pattern in the central city vs. outer zones and the subsequent variation on the dynamics of the UGS. Besides, earlier spatial-temporal studies on UGS dynamics in Addis Ababa has reported a mixed evidence. Some of these studies have stated that there has been a consistent decline of UGS [14–17], while others identified that there was an increase in the proportion of the UGS [4], displaying inconsistencies in the studies. Hence, understanding the actual magnitude and direction of change will deepen our scientific understanding and provide policy options for imperative urban planning intervention measures.

Remote Sensing (RS) and Geographical Information System (GIS) techniques has widely been used to investigate spatio-temporal dynamics of urban expansion and UGS change. Over the past couple of decades, the advancement of RS and GIS has provided opportunity for a better understanding of Land Use Land Cover (LULC) changes. Literatures have shown that the LULC studies have been conducted at various spatial scales, i.e., global [3,12,19], regional [20–22] sub-regional [6,23,24], city scale [14,25], multiple cities [26–28], sub-city [13] or a district scale [29]. Large scale or city wide LULC analysis sometimes fails to uncover spatial variation of urban dynamism and UGS change because spatial variabilities of change may not easily be observed. Besides, the conventional RS and GIS has limitation in quantifying the changes because it lacks indices for quantifying urban dynamism in two or more time points [30]. Therefore, new sources of spatial data and innovative techniques that offer the potential to significantly improve the analysis, understanding, presentation and modelling of urban dynamism [31], is needed to measure

urban expansion and UGS change. As a result, there has been an increasing interest in employing the Landscape Expansion Index (LEI) to measure urban growth patterns in recent years [30]. One of the advantages of the LEI over the RS and GIS techniques is that it has the capability to capture complex urban growth using multi-temporal remote sensing data [30]. Hence, investigating such a complex process using zonal and directional analysis will provide a new insight on the spatial variabilities of urban expansion and UGS change in the city.

This study used RS and GIS technology integrated with LEI to monitor urban expansion and UGS change in the inner zone, outer zone and eight directions in Addis Ababa, Ethiopia. It was aimed to assess the magnitude, directions of urban expansion and UGS change as well as spatial variations. The UGS in this study refers to vegetation, urban forest, agricultural land, grass, parks, wood lots and enclosed garden spaces that found in private and public compounds, and exclude non-vegetated spaces, as indicated in the Ethiopian national urban green infrastructure standard and as used in earlier studies [4,14,32].

## 2. Methodology

### 2.1. Study Area

Addis Abba is the capital of Ethiopia and is located at 9°0′19.4436″ N and 38°45′48.9996″ E. It is the largest urban center in the country and hosts almost one-fourth of the urban population in the country [33]. The city's population size rapidly expanded from 1.4 million in 1984 to 3.4 million in 2019 [34,35], increased more than two fold (142.8%). Administratively, the city is divided into ten sub-cities and 116 districts (woredas). It covers an area of 520 km$^2$ [36]. The city was established as the nation's capital in 1886, which makes it a relatively younger city compared to many other African cities [4]. Since its establishment, for nearly a century, the city didn't exhibited significant urban landscape transformation because of land tenure system and restrictive economic policy imposed during the military government. However, after 1991 with the change of government and the subsequent shift from a socialist economic policy to market-oriented policy, the city demonstrated rapid urban landscape transformation [14]. The transformation process, however, accompanied by mainly unplanned urban expansion and a substantial decline of UGS [13,14,16,17]. Arguably, Abebe and Megento [13] in their study on Bole sub city in Addis Ababa from a "Forest city" to "Urban Heat Island" revealed that the city is almost transformed into urban habitats due destruction the UGS and uncoordinated development. Similarly, Zewdie and Worku [14] reported that the spatial extent of the city doubled over the past three decades. In spite of this, in recent years studies have shown that there was an increase in the UGS proportion in central city areas, while there was a decline of UGS in the outer zone [4]. However, spatial variations across the city's landscape and actual magnitude of change remained unexplored as the earlier studies were conducted at a larger scale and utilized conventional spatio-temporal dynamic analysis methods, as mentioned earlier.

### 2.2. Sources of Data

The study used remote sensing imagery for Addis Ababa for different times in 1989, 1999, 2009 and 2019. These reference years were selected to divide the time into the decade intervals and to represent the establishment of the first National Urban Planning Institute with proclamation No. 317/87 in the country's history. Prior to 1987, guiding urban centers through spatial plans had never been practiced (with exception of Addis Ababa) because the urbanization level of the country was very low (8%) [34]. Hence, understanding how country's urban centers such as Addis Ababa have been expanding over the past decades provides scientific evidence on the trend of urbanization and the impact on the UGS. In addition, in Ethiopia Land Use Land Cover (LULC) data that offers historical information on urban dynamics that rarely exist, because spatio-temporal inquiries are at its infancy. None of such studies used hierarchical (multiscale) approach in measuring urban expansion and UGS change. Therefore, analyzing the changes using medium resolution

satellite images will expand our knowledge on the urbanization pattern and UGS change in the city.

In this study, images from Landsat Multispectral Scanner (MSS), Thematic Mapper Plus (TM), Enhanced Thematic Mapper Plus (ETM+), and Operational Land Imager (OLI_TIRS), all of required spatial resolution between 60 m and 30 m, were obtained from the Global Land Cover Facility (GLCF). However, as the 1989 image had a 60 m resolution it was resampled to 30 m to ensure data consistency for the analysis (Table 1). The images were collected from dry seasons to reduce the availability of haze and cloud-free satellite images and enhance interpretation and LULC change. As the pre-image processing step, atmospheric and radiometric correction was carried out before classification. The process was followed by the projection of the images to a common coordinate system called the Universal Transverse Mercator (UTM) of WGS84 and Datum Zone 37.

**Table 1.** Satellite image source used for the study and their spatial resolution.

| No | Satellite | Sensor ID | Resolution | Date Acquired | Path & Raw | Source |
|----|-----------|-----------|------------|---------------|------------|--------|
| 1 | Landsat 4 | MSS | 60 m | 21-December-1985 | 169/055 | Global Land |
| 2 | Landsat 5 | TM | 30 m | 31-January-2019 | 169/055 | Cover Facility |
| 3 | Landsat 7 | ETM+ | 30 m | 15-December-2017 | 169/055 | www.glcf.umiacs.umd.edu |
| 4 | Landsat 8 | OLI_TIRS | 30 m | 20-December-2016 | 169/055 | (accessed date 12 February 2019) |

The LULC classification employed the two most common classification methods, i.e., supervised and unsupervised classifications. First, unsupervised image classification was carried to determine strata for ground truth. Then the supervised classification was carried out using a Maximum Likelihood Classifier (MLC), which is the typical method that creates a decision surface-based mean and covariance of each class [37]. Eventually, six LULC classes such as agriculture, built up, bare soil, urban green/vegetation, urban forest and water body, were identified as shown below in the Table 2. The classification achieved an accuracy of over 85% after evaluation against the latest master plan, 300 training samples and a high-resolution Google Earth Map. As the objective of the study was to monitor urban expansion and UGS change, the identified LULC were categorized into two classes. The built-up area and bare soil were categorized as urban as suggested by [4], while agriculture, urban green/vegetation and urban forest were classified as UGS, as suggested by Ethiopian National Urban Green Infrastructure Standard and previous studies [13,14,32].

**Table 2.** Land use/cover classification scheme employed for the study.

| Land Use/Land Cover | Description/Definition | Key Authors |
|---------------------|------------------------|-------------|
| Agriculture | Cropland, pasture and grassland. | [13,14] |
| Built up | Residential, commercial services, industrial area, transportation, communications, and utilities mixed urban or other built-up areas. | [13,14] |
| Bare soil | Parts of the land surface which are mainly covered by bare soil | [13] |
| Urban green/vegetation | Any vegetation found in the urban environment, including parks, residential gardens or street trees | [13,14] |
| Natural forest | Natural and plantation forest, trees, and other woody species. | [13,14] |
| Water body | Rivers, small streams, ponds, etc. | [36] |

Finally, newly grown urban patches between two times periods were detected by spatially overlying two temporally adjacent maps (1989–1999, 1999–2009 and 2009–2019) to identify the pattern of growth, e.g., infill, edge expansion, and an outlying.

*2.3. Urban Growth and Urban Green Space Change Analysis*

Urban Expansion (UE) was used to measure the magnitude and direction of urban growth [26], and it was computed as follows:

$$UE = Ue - Ui/T \tag{1}$$

where Ue and Ui represent urban extent at the initial and end of the monitoring period, respectively, and T is the period from the time e to i. Similarly, the annual growth rate which compares different geographical areas of a city in terms of the [26] was utilized for analyzing the pace of urbanization. The annual rate of change was calculated using the following formula:

$$r = \frac{1}{t_2 - t_1}\left(\ln\frac{A_{t2}}{A_{t1}}\right) \tag{2}$$

where $A_{t2}$ and $A_{t1}$ are the built up land area in the year $t_2$ and year $t_1$, respectively. This equation has been widely used to calculate the annual growth rate because the method assumes urban growth is an exponential to the annual rate of compound interest [38].

Similarly, the Urbanization Intensity Index (UII) index, which is the ratio of the area of urban land expansion to the total land area in a spatial unit in the study period [24], was employed to compare the pace and intensity of urban expansion in the inner zone, outer zone and eight directions. The advantage of UII is that it normalizes the annual mean expansion rate based on the land area in a spatial unit, thereby enabling comparative analysis [24]. It was computed using the following formula:

$$UII = U_a - U_b/T \times U_c \times 100\% \tag{3}$$

where UII is the expansion intensity in the ith spatial unit, $U_a$ is the area of the urban land in the ith spatial unit in period a, $U_b$ is the area of the urban land in the ith spatial unit in period b, $U_c$ is the total land area of the ith spatial unit, and T is the time span from period a to period b in the unit of year.

In addition, Urban Growth Coefficient (UGC) was used to calculate whether the urban growth is sprawling or densifying as used by [39]. The UGC was computed as follows:

$$UGC = \text{Rate of Urban Expansion}/\text{Rate of Urban Population Growth} \tag{4}$$

According to Rode and Heeckt [39], a UGC value greater than 1 indicates sprawling, implying the built-up land is increasing faster than the population growth. A UGC value of less than 1 means that the city is exhibiting densification or compact development.

Moreover, the LEI was utilized to analyze urban expansion patterns in Addis Ababa between 1989 and 2019, as indicated by [30]. The advantage of LEI is that it captures information on the processes of a landscape pattern [30]. Importantly, the LEI illustrates the different modes of urban expansion for new urban patches [30], e.g., infilling, edge expansion and outlying. The LEI of each new urban patch was calculated year by year using the formula:

$$LEI = \frac{A_0}{A_0 + A_v} \times 100 \tag{5}$$

Where $A_0$ is the intersection between the buffer around a new urban patch and the previously existing urban land, and Av is the intersection between the buffer zone and the previously non-green urban area. Based on the result of the LEI, urban growth can be classified into three modes: infilling, edge expansion and outlying [30], as mentioned above. The infilling mode of urban growth refers to when the growth occurs between old urban patches and new urban patches (i.e., LEI is between 50 and 100). The edge expansion mode of urban growth is when a new urban patch expands from the edges of an existing urban patch (i.e., LEI is between 0 and 50). The leapfrog mode of urban growth is when a new urban patch is isolated from the old ones (i.e., LEI is equal to 0) [30].

In addition, the change in UGS is also measured by using the greenness index. The greenness index gives information on the environmental quality of cities and the proportion of green spaces in relation to urban land cover types [40]. High green index means high proportion of UGS, low green index means low proportion of the UGS in the study area.

In this study, the green index method of estimation, proposed by [40] was utilized and calculated as the percentage of the total green area divided by the total size of the urban area.

$$\text{Greenness Index} \; = \; \frac{\text{Area Covered by green}}{\text{Total area of study area}} \tag{6}$$

### 2.4. Directional and Zonal Analysis

Linear gradient and zonal analysis have recently been widely used for quantifying Spatio-temporal dynamics of urban expansion and change in UGS [22,41,42]. However, the linear gradient analysis leads to a bias towards the investigation of urban land use [43], because it utilizes concentric circles for analyzing distance decay effects. Added to that, gradient analysis has limitations in characterizing the complexity of urban form morphology and the diversity of deriving forces, especially in metropolitan areas where the regional urbanization process is not only determined by urban growth of the central city but also shaped by rural urbanization [31].

Owing to this limitations, in this study, we used zonal and directional analysis to measure urban expansion and UGS change. The advantage of directional analysis is that it can help establish spatial relationships between land cover changes and dynamics from a city center [41] and such research has seldom been documented so far [31]. Thus, in this study we utilized combined directional and zonal study, as shown in Figure 1. Referring Figure 1, Addis Ababa city is divided into ten sub-cities of which four sub-cities (namely, Addis Ketema, Arada, Lideta and Kirkos) are located in the inner urban core zone. We considered these sub-cities as inner zone because the sub-cities are located in central areas and represent the oldest parts of the city. The other six sub-cities were categorized as outer zone, because they are sharing boundaries with a neighboring region. In addition, the study area was divided into eight quadrants at an angle $45^0$ based on road availability for measuring variations of urbanization and UGS change in the study area, as shown in Figure 1. The location of the city administration office, which represents the first settlement of the area, was used as a center for directional analysis. The eight quadrants were South South East (SSE), South East East (SEE), South South West (SSW), South West West (SWW), North West West (NWW), North North West (NNW), North North East (NNE), North East East (NEE).

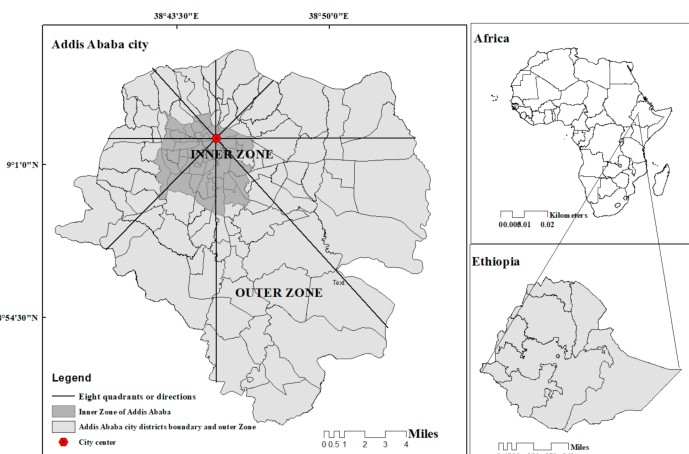

**Figure 1.** Addis Ababa urban core zone, inner urban zone and peri-urban zone and eight directions.

Besides, as the study focused on how the urban area shifted in parallel with urbanization, it was necessary to measure the change of the center of gravity. Hence, the geometric center of gravity of urban area distribution in the city was measured to reflect the overall

trend of urban expansion in Addis Ababa. It was computed using the following formula as suggested by [44]:

$$\text{Gravity center of urban area UA } (x,y) \quad \frac{\sum_i^n U_i Q(X_i y_i)}{\sum_i^n U} \qquad (7)$$

where UA (*x*,*y*) are the geographic coordinates of the center of gravity of urban area in Addis Ababa, Ui is the area of urban patches of respective years, and Q (*x*,*y*) are the geographic coordinates of the center of gravity of urban patches in the city.

### 3. Result

#### *3.1. Spatio-Temporal Dynamics of Urban Expansion and Urban Green Spaces Change*

From Figure 2, the urban area in the study area increased from 14,400 ha to 18,840 ha (30.3%), from 18,840 ha to 25,250 ha (34%) and from 25,250 ha to 31,900 ha (26.3%) in the first period (1989 to 1999), the second period (1999 to 2009), and third periods (2009 to 2019), respectively, with an annual rate of increase 12.1ha/per year. The urban growth was highest in the second period compared to the first and third periods, indicating urbanization intensified in this period compared to others. Over the past three decades, the urban extent of Addis Ababa increased more than two-fold i.e., increasing from 14,400 ha to 31,900 ha (121.5%), and demonstrating intensive and rapid urbanization.

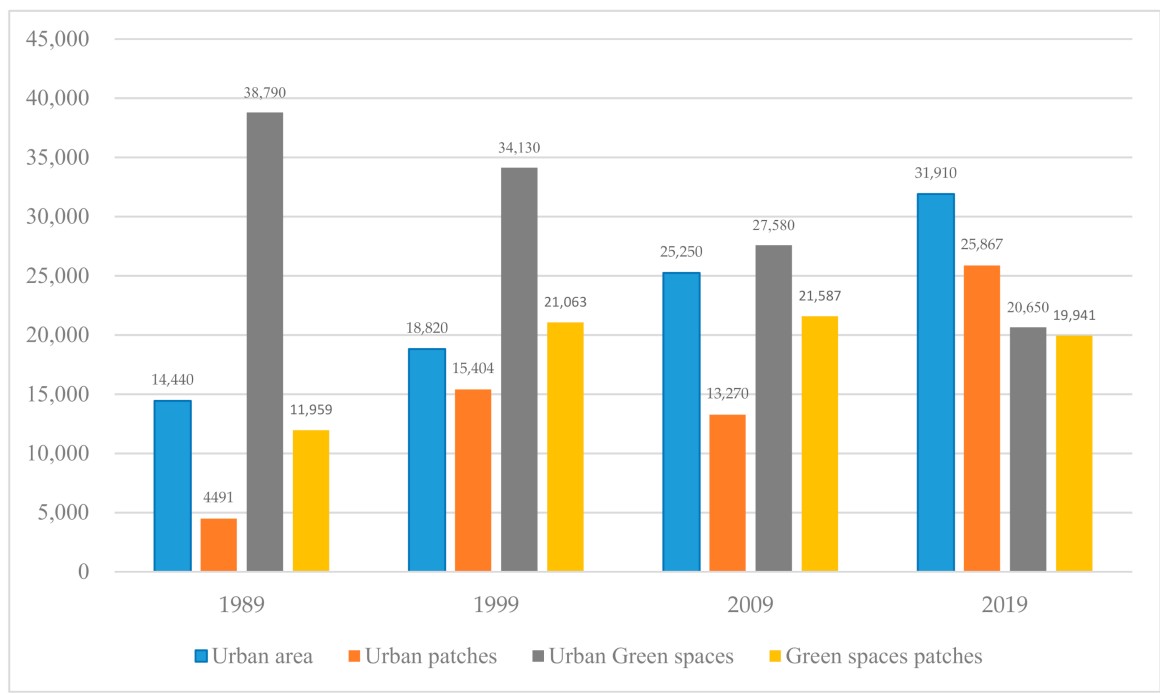

**Figure 2.** Urban expansion and urban green spaces change in Addis Ababa from1989 to 2019.

Similarly, the results also showed that the number of urban patches (NP) generally increased but fluctuated in parallel with the urban expansion, i.e., increased and decreased in the study periods. Referring to Figure 2, the highest NP increase was observed in the third period compared to the first and the second period. For instance, in 1989, the NP of urban patches were 4491 and this number increased to 15,404 (242.9%) by 1999, while it decreased from 15,404 to 13,270 (13.8%) between 1999 and 2009. In the third period, however, the NP further increased (nearly doubled) from 13,270 to 25,867 (94.7%), showing differentiation and isolation of urban patches i.e., forming a non-contagious patches. The increase in the number of urban patches in this period suggested that the pattern of urban growth mainly characterized dispersion or spontaneous development.

On the other hand, the study also showed that the UGS significantly declined in the study period, in parallel with rapid urban expansion. Illustratively, the UGS declined from 38,790 ha to 34,390 ha (12%), 34,390 to 27,380 and (19%) and from 27,380 to 20,650 ha (25.1%) with reduction rate of 1.1%, 2%, and 2.4% in the first, second, and third periods, respectively. The reduction rate of the UGS in the third period was twice as high as the rate in the first period, revealing increasing trend of UGS due to urban expansion accompanied by low density development. Correspondingly, the dynamics of NP for the UGS provided evidence on the dynamics. The NP for the UGS exhibited a varying trend of change during the study period. Referring to Figure 2, the NP increased from 11,959 to 21,063 (76.2%) in the first period, showing spontaneous urban growth characterized by the fragmentation of UGS patches. While in the second period the NP increased marginally from 21,060 to 21,587 (2.4%). The slight increase in this period shows that the city mainly focused on infilling or edge expansion, which is a typical pattern of urban growth when a city pursues compact urban growth. Similarly, in the third period, NP further declined from 21,587 to 19,942 (7.6%), exhibiting an increasing trend of conversion of existing UGS patches to urban functional uses.

In addition, the result of the green index (GI) provided similar evidence on the steady decline of UGS in the city. As shown in Figure 3, the GI was 68% in 1989 and this proportion declined to 63% in 1999 and further reduced to 51% in 2009, reporting fast destruction of the UGS over the study period. By the end of 2019, the GI of the city decreased to 38% and shared less than half the city's urban landscape, witnessing rapid depletion of the UGS. Similarly, the rate of GI change was 0.7, 1.9 and 2.4 per year in the first, second and third periods and demonstrating growing trend of UGS destruction over the study period due to uncontrolled urban growth.

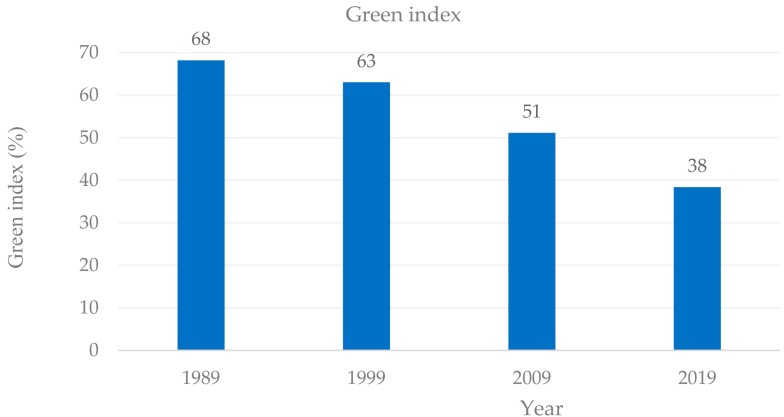

**Figure 3.** The ratio of Urban Green Spaces to total area and built-up area in Addis Ababa.

Looking at Table 3, the UII consistently increased in the study periods. For example, the UII reported 0.81, 1.19 and 1.28 in the first, second, and third periods, respectively, presenting the increasing transformation of the urban landscape in the city. In addition, comparison of urban growth coefficient and population growth rate (UGC) provided a value greater than 1 in the second and third period, showing that the city predominately exhibited urban sprawl, i.e., a low-density growth characterized by dispersed urban patches (Table 3). The growth, however, more intensified in the second period (1999 to 2009), compared to the first (1989 to 1999) and third period (2009 to 2019). The highest UGC value in the second period indicates that urban growth in this period more intensified than compared to other periods. Contrarily, the UGC value in the first period was less than 1, indicating urban expansion was slow compared to population growth. This period marks the socialist period where private sector involvement in the economy was limited, because the government had controlled solely investments and this policy reduced demand for urban land.

**Table 3.** Showing population size, growth rate, urbanization intensity index, annual expansion and urban growth coefficient of Addis Ababa from 1989 to 2019.

| Year | Population (million) | Growth Rate (%) | Urbanization Intensity Index (UEI) | Annual Expansion Rate (AER) | Urban Growth Coefficient (UGC) |
|------|------|------|------|------|------|
| 1989 | 1.4 | - | - | - | - |
| 1999 | 2.1 | 4.7 | 0.81 | 3.03 | 0.64 |
| 2009 | 2.7 | 2.8 | 1.19 | 3.41 | 1.22 |
| 2019 | 3.4 | 2.6 | 1.28 | 2.63 | 1.01 |

### 3.2. Urban Growth Pattern in Addis Ababa from 1989 to 2019

The LEI analysis showed that the city displayed an outlying urban growth pattern signficantly. From Figure 4, an outlying growth accounted for more than 98.9% of the spatial growth, showing that the urban expansion in the study area was characterized by the differentiation of urban patches. An outlying growth was identified as the most dynamic urban growth pattern and increased from 15,130 ha in the first period to 21,330 ha in the second period, and further increased to 28,270 ha in the third period, reporting spontaneous urban growth pattern.

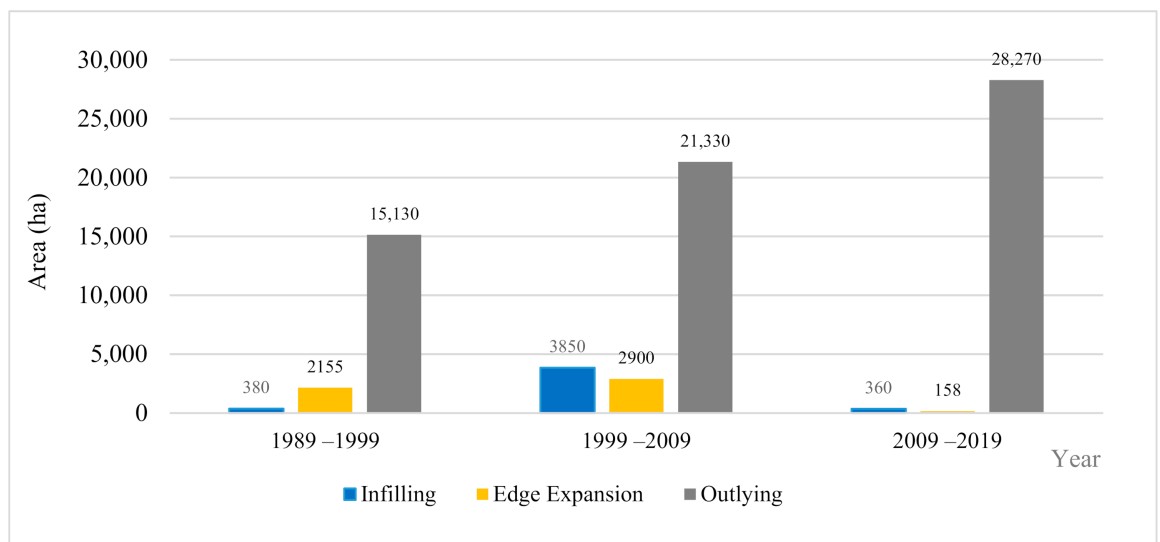

**Figure 4.** Urban growth pattern in Addis Ababa from 1989 to 2019.

Meanwhile, edge expansion and infilling growth accounted for an insignificant share compared to the outlying growth (Figure 4). For example, the infilling growth pattern accounted for 380 ha (0.024%), 3850 ha (0.1771%) and 360 ha (0.012%) in the first, second, and third periods, respectively. The infill growth pattern increased slightly in the second period but decreased in the first and third periods. The declining trend of infilling growth in the third period may attributed to the fact that the existing UGS and open spaces were converted to functional urban uses and there were no more patches to further infill growth to occur. Similarly, edge expansion, which was the second type of urban expansion pattern, showed 2150 ha (1.4%), 2900 ha (1.33%), and 158 ha (0.05%), in the first, second, and third periods, respectively. Unlike the infill growth, the edge expansion pattern growth demonstrated a relatively higher share of urban growth pattern. The higher edge expansion in the first period compared to the second period implies that there was a compact urban growth policy implemented in the period. In addition, the number of patches (NP) exhibited changes over the study period. The number of newly grown patches with infilling expansion was 15,101 and 10, which corresponds to 0.14%, 1.57% and 0.145%, respectively. The NP with the edge expansion pattern was 111, 429 and 22, which equates to 0.037%, 6.77%, and 0.37%, respectively. On the other hand, the NP of the newly

grown outlying patches pattern was 10,575, 5867 and 6831, which corresponds to 98.8%, 91.7% and 99.5%, respectively. (Table 4).

**Table 4.** The number of patches in growth pattern types and their proportion in Addis Ababa from 1989 to 2019.

| Landscape Expansion Index Pattern | 1989–1999 | | 1999–2009 | | 2009–2019 | |
|---|---|---|---|---|---|---|
| | NP | Proportion (%) | NP | Proportion (%) | NP | Proportion (%) |
| Infilling | 15 | 0.140 | 101 | 1.57 | 10 | 0.145 |
| Edge expansion | 111 | 0.037 | 429 | 6.70 | 22 | 0.32 |
| Outlying | 10,575 | 98.82 | 5867 | 91.7 | 6831 | 99.53 |

### 3.3. Urban Expansion and Urban Green Spaces Change in Inner and Outer Zone

The spatial-temporal patterns of urban expansion and UGS change were analyzed in the inner zone and outer zone to understand variation in their dynamism (Table 5). Results that two zones exhibited a varying degree of change from 1989 to 2019. For instance, urban growth in the inner zone increased in the first period (from 3712 ha to 3716 ha) and second period (from 3716 ha to 3874 ha), while it decreased in the third period (from 3874 ha to 3733 ha). The increase in the urban area in the inner zone in the first period was insignificant (0.39%), while it was substantial (15.81%) in the second period. Conversely, the UGS in the inner zone declined in the first period (from 60 ha to 54 ha) and the second period (from 54 ha to 38 ha), while it increased in the third period (from 38 ha to 53 ha) (Table 5). The increase in the UGS proportion in the inner zone was significant in the third period (increased by 15%), reporting there was an effort to in increase the quantity of the UGS in this period through urban planning interventions measures. Similarly, urban expansion in the outer zone increased from 10,729 ha to 15,112 ha, from 15,112 ha to 21,377 ha, and from 21,377 ha to 28,176 ha in the first, second and third period with an annual rate of expansion of 4% 4.1% and 3.1%, respectively.

**Table 5.** The proportion number of patches based on urban growth types in Addis Ababa from 1989 to 2019.

| Year | Land Cover Area (ha) | | | | Change Rate ha/year) | | | | Urbanization Intensity Index | |
|---|---|---|---|---|---|---|---|---|---|---|
| | Urban (ha) | | UGS (ha) | | Urban | | UGS | | | |
| | Inner Zone | Outer Zone | Inner Zone | Outer Zone | Inner Zone | Outer Zone | Inner Zone | Outer Zone | Inner Zone | Outer Zone |
| 1989 | 3712 | 10,729 | 60 | 3624 | - | - | - | - | - | - |
| 1999 | 3716 | 15,112 | 54 | 3171 | 0.1 | 4 | −1 | −1.2 | 0.009 | 0.88 |
| 2009 | 3874 | 21,377 | 38 | 2555 | 0.42 | 4.1 | −2.9 | −1.8 | 0.35 | 1.24 |
| 2019 | 3733 | 28,176 | 53 | 1879 | 0.36 | 3.1 | 3.9 | −2.6 | −0.34 | 1 |

Unlike the inner zone, the UGS in the outer zone demonstrated a persistent decline during the study period. The UGS in the outer zone decreased from 3624 ha to 3121 ha, from 3171ha to 2555 ha and from 2550 ha to 1879 ha in the first, second and third study periods. The rate of UGS decline in the first, second and third periods corresponds to 1.2%, 1.8% and 2.6%, respectively. The rate of UGS decline in the third period was twice as high as the first period indicating UGS destruction intensified in the third period.

### 3.4. Directional Analysis of Urban Expansion and Urban Green Spaces Change in Addis Ababa from 1989 to 2019

The spatio-temporal dynamics of urban expansion and the UGS change were also analyzed for eight different directions using equal-fan analysis. Referring to Figure 5 urban expansion demonstrated distinct magnitude and direction change in eight directions over the 30 years. The urban area mainly expanded in the SEE, SSE, and SSW directions from

3026 ha to 4650 ha, from 4990 ha to 5337 ha and from 2891 ha to 3622 ha, respectively, at the beginning of the study period. In the second period, however, urban growth shifted to SEE, NEE and SSW and increased from 4650 ha to 6154 ha, from 1140 ha to 2098 ha, and from 3622 ha to 4672 ha, with an annual rate expansion of 4.78%, 3.62% and 2.85%, respectively. As urbanization intensified in the third period, the urban expansion further shifted to the SSE and NEE directions and increased from 6145 ha to 9365 ha and from 2098 ha to 2011 ha, respectively. During this period, urban expansion in the SEE direction was maintained and increased from 6154 ha to 8378 ha. The rate of urban expansion in SEE, NEE and SEE was 6.03%, 3.62%, and 4.78%, respectively, reporting the city expanded mainly in the SEE directions during the study period.

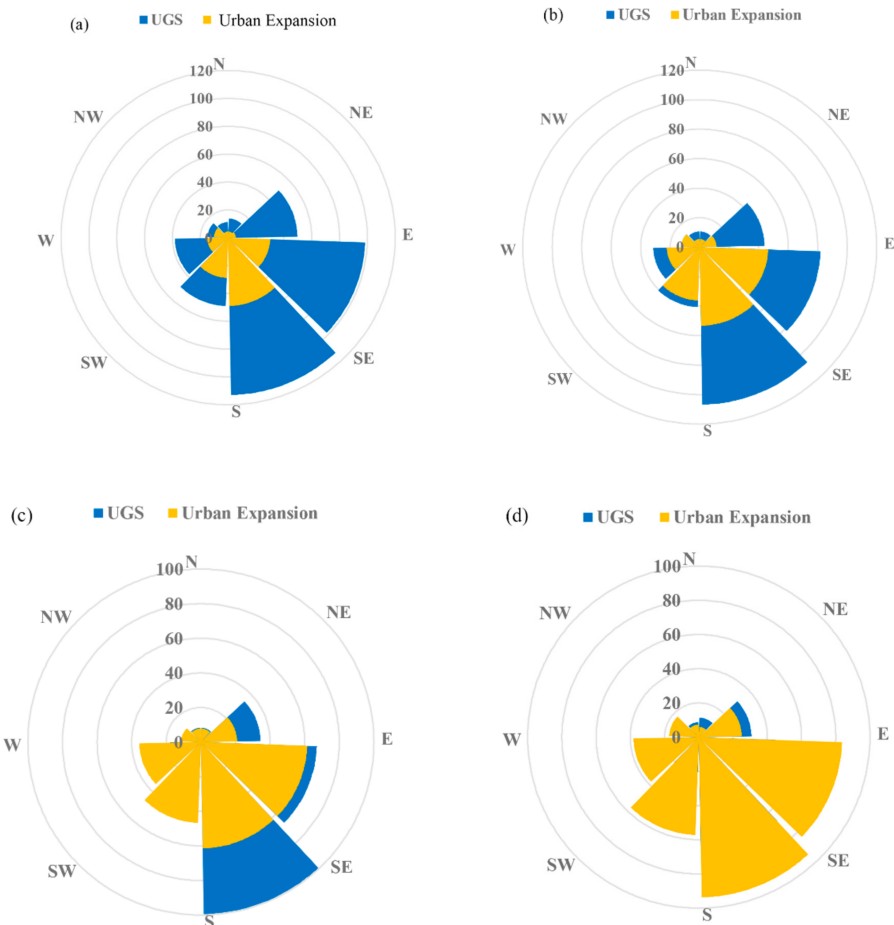

**Figure 5.** Directional analysis of urban expansion and urban green space change in eight quadrants: (**a**) in 1989, (**b**) in 1999, (**c**) in 2009, (**d**) in 2019.

Importantly, one of the noticeable findings of the study was that urban area displayed a decreased in some quadrants. For example, the urban area increased in the first and second period in all quadrants but decreased in the third period in NWW, NNW, and NNE directions from 1756 ha to 1732 ha, from 785 ha to 729 ha and from 726 ha to 603 ha with a rate of 0.19%, 0.84%, and 2.95%, respectively.

The low level of urban expansion in the NWW, NNW and NNE direction was associated with the natural characteristics of the study area. The spatial characteristics of the areas acted as barriers for urban expansion. Unlike the urban expansion, the computed result for the UGS dynamism showed varying change in the NWW, NNW and NNE directions. For instance, the UGS coverage in these directions decreased in the first and second period but increased in the third period. The UGS growth in the NWW, NNW, and NNE directions in the third period was 215 ha 456 ha and 1380 ha, respectively (Table 6) and this may

attributed to the recent summer tree plantation campaigns and the presence of protected botanic garden.

**Table 6.** Urban expansion and urban green space dynamics in eight quadrants in Addis Ababa.

| Land Cover | Direction | Area (Ha) | Urban Expansion in Eight Directions (Ha) | | | | Change Rate (Ha/year) | | | Change (Ha) | Intensity |
|---|---|---|---|---|---|---|---|---|---|---|---|
| | | | 1989 | 1999 | 2009 | 2019 | (1989–1999) | (1999–2009) | (2009–2019) | (1989–2019) | |
| Urban | SSE | 16,260 | 4909 | 5337 | 6145 | 9365 | 0.26 | 1.65 | 6.03 | 4456 | 2.74 |
| UGS | | | 11300 | 10684 | 9943 | 6738 | −0.38 | −0.66 | −3.00 | −4562 | |
| Urban | SEE | 12,960 | 3026 | 4650 | 6154 | 8378 | 1.25 | 4.97 | 4.78 | 5352 | 4.13 |
| UGS | | | 9839 | 8195 | 6707 | 4493 | −1.27 | −1.51 | −2.70 | −5346 | |
| Urban | SSW | 7840 | 2891 | 3622 | 4672 | 5719 | 0.93 | 3.63 | 2.89 | 2828 | 3.61 |
| UGS | | | 4911 | 4064 | 3088 | 2032 | −1.08 | −1.99 | −2.60 | −2879 | |
| Urban | SWW | 5450 | 1507 | 2234 | 3559 | 3817 | 1.33 | 8.79 | 1.15 | 2310 | 4.24 |
| UGS | | | 3826 | 3147 | 1801 | 1517 | −1.25 | −3.52 | −0.90 | −2309 | |
| Urban | NWW | 2560 | 1024 | 1262 | 1756 | 1732 | 0.93 | 4.82 | −0.19 | 708 | 2.77 |
| UGS | | | 1474 | 1242 | 721 | 743 | −0.91 | −3.53 | 0.18 | −731 | |
| Urban | NNW | 1630 | 443 | 549 | 785 | 739 | 0.65 | 5.33 | −0.84 | 296 | 1.82 |
| UGS | | | 1120 | 1061 | 833 | 879 | −0.36 | −2.04 | 0.43 | −241 | |
| Urban | NNE | 1780 | 320 | 417 | 726 | 603 | 0.54 | 9.66 | −2.95 | 283 | 1.59 |
| UGS | | | 1380 | 1061 | 1000 | 1138 | −1.79 | −0.44 | 1.30 | −242 | |
| Urban | NEE | 5680 | 553 | 1140 | 2098 | 2511 | 1.03 | 17.32 | 3.62 | 1958 | 3.45 |
| UGS | | | 4970 | 4390 | 3446 | 3071 | −1.02 | −1.90 | −0.85 | −1899 | |

Added to that, Table 6 depicts the spatial variations and distribution of UII in eight quadrants based on the urban expansion in the study area. The result revealed that the highest UII was reported in the SEE, SWW and SSW directions. These areas have relatively flat topography and accessible due to expanded road network infrastructure, which is suitable for urban expansion.

The NEE, NWW and SSE directions also showed high UII and urbanization, despite the fact that some of these quadrants are unsuitable for urban expansion due to their physical characteristics. In other words, the result unfolded that urban expansion have been observed in Addis Ababa beyond the permissible range slope for urban construction. This implies that unsuitable urban landscape areas for urban construction such as hilly areas which are usually recommended for urban environmental function in spatial plans have been engulfed by built-up functions, which witnesses the city is not pursing sustainable urban growth.

In conclusion, while the city was experiencing rapid urban expansion the center of gravity is also changed over the study period. Referring to Figure 6 the geometric center of gravity UA generally moved northward and then southward during the study period. In the first period, the center of gravity gradually moved to the northeast direction and continued until the end of the second period. However, in the third period, the center of gravity shifted to southward directions, showing the large distance between the geometric centers of UA compared to the first and second periods. The change in the center of gravity to the southern direction indicates that urbanization more pronounced in this direction, because of urbanization inducing factors such as suitability of the land and accessibility, expansion of private and public investments.

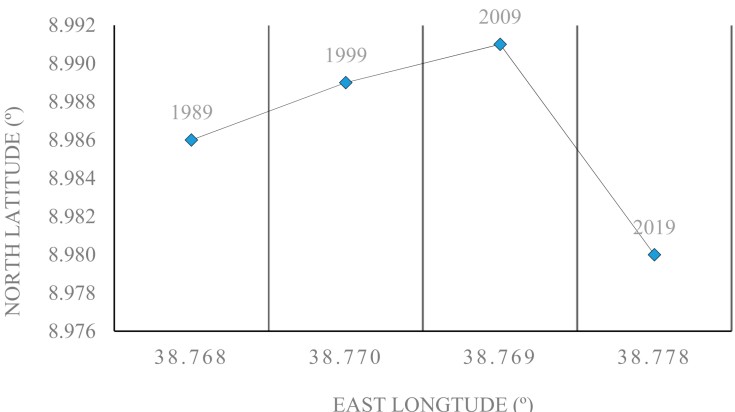

**Figure 6.** Center of gravity of dynamic change in urban area in Addis Ababa from 1989 to 2019.

## 4. Discussion

The present study demonstrated that the spatio-temporal pattern of urban expansion and UGS change in Addis Ababa using directional and zonal analysis from 1989 to 2019. The result showed that the urban area progressively expanded in the city, demonstrating intensive and rapid urbanization observed over 30 years. Similar studies have been reported that Addis Ababa experienced rapid urban expansion and reduction of UGS due to uncontrolled and uncoordinated urbanization [13,14,16]. For instance, Abebe and Megento [13] identified that the urban area in Addis Ababa increased with an annual rate of 5.7% and consumed 419% of the city's total area (mostly UGS) between 1986 and 2015. Correspondingly, Zewdie and Worku [14] reported that the urban area in Addis Ababa expanded by 50%, while agricultural land and forest decreased by 34% and 16% between 1984 and 2014. In agreement with this finding, other studies carried out in similar cities in the African continent reported that there has been rapid urban expansion and UGS decline due to unprecedented urbanization. For example, Gruter [45] identified that there was a substantial reductions and fragmentations in agricultural lands, riverine and open forest, while there has been over 200 percent increase in built-up areas in metropolitan Accra, Ghana. Unlike the above studies, this study indicated that the city exhibited a varying magnitude of urban expansion in the inner zone, outer zone and in eight directions. According to this study, the urban area consistently expanded in the outer zone, indicating continued suburbanization, while it demonstrated a mixed direction in the inner zone (increased in the first period and the second period, but decreased in the third period). Obviously, the urban expansion in Addis Ababa different from the urban expansion observed in Accra [46] and expansion reported in Chinese cities along the Yangtze river in the same periods [24,47]. In the Metropolitan Accra, urban expansion in the historical origin of the urban core accounted for more than half of the total built-up land increased over the 28-year period [46]. On a similar account, Sun and Shan [24] in their studies in Chinese cities concluded that there was a significant increase in built-up areas along the Yangtze River with rapid expansion in the urban core, demonstrating that Addis Ababa pursued different direction of urban expansion compared to Accra and Chinese cities. Similarly, Xue and Hou [48] in their investigation on quantifying spatiotemporal dynamics of urban growth modes in metropolitan cities of China including Beijing, Shanghai, Tianjin and Guangzhou, identified that in Beijing China the prominent urban growth occurred around the city core with increasing contribution of edge expansion in a similar period. Variation in the pattern of urban expansion in Beijing and Addis Ababa may stem from urban planning measures implemented in Chinese cities compared to Addis Ababa. Chinese cities effectively enforce and implement urban planning intervention measures towards a coordinated urban development [42].

Importantly, the study also showed that like many other African cities, urban expansion in Addis Ababa accompanied by reductions of the UGS, a finding which compares with earlier studies [13,14,16,49]. Lindley and Gill [50] concluded that urbanization in

African country cities is characterized by the conversion of UGS, which is referred to as grey extensification [12]. According to Richards and Belcher [12], urban expansion in developing country cities in Africa, Asia, and South America was characterized by a rapid decline of vegetation cover or UGS; while urbanization in some European countries demonstrated a 10% increased between 2000 and 2015 [12]. On contrary, the current study showed that there was an increase in the proportion of the UGS in the inner zone through reduction of urban area, displaying a spatial trade-off between the two main urban land use functions. Illustratively, in the third period, while the city was expanding outward, massive urban redevelopment projects undertook in the inner zone to improve the city's competitiveness as a business location, to tackle the huge backlog in affordable housing and basic service delivery through accelerated investment in infrastructure and public housing programs [26]. This, along with subsequent introduction of new urban planning regulations, standards and norms for new construction and redevelopment projects as well as allocation new parks such as (Economic Commission for Africa) ECA, significantly contributed to increases in the proportion of UGS in the inner zone. In line with this, a similar study carried out in secondary towns in Ethiopia such as Hawassa and Bahirdar revealed that the proportion of green spaces increased with increasing urbanization [51]. Similar investigations have also been reported in Asian countries. For instance, Zhou and Wang [42] concluded that the proportion of UGS can increase with urban growth, if strong measures are implemented to protect natural resources. Correspondingly, Haasa and Furberg [52] argued that with an increase in the built-up area, the UGS in Shanghai, Stockholm and Kualalaumpur, increased by 318%, 110% and 425%, respectively, showing that UGS can increased with urban growth if protection to existing green spaces and allocation of new ones are done. Therefore, in general, in agreement with the study [42,51], the findings suggest that urban growth does not necessarily result in the decline of UGS, if due planning measures are taken to preserve urban green spaces even in African context.

On the other hand, the directional analysis showed that Addis Ababa demonstrated distinct direction of urban expansion over the past three decades. According to the UII score, SEE,> SSW,>SSW,> SWW,> and NEE directions reported greater value, which implies that there was a high-speed urban expansion in these directions compared to the others. The rapid expansion in these directions may be attributed to the size of the sector, relatively flat terrain of the quadrants, suitability of land for urban construction, increasing accessibility due to expansion of high ways, development of public projects such as IHDP, expansion of private industries and industrial parks [14,26]. Obviously, It was identified that 13,964.8 ha (85.4%) of urban expansion occurred in these four quadrants, a result which resonates with similar studies carried out in the city. For instance, Terfa and Chen [26] and Zewdie and Worku [14] identified that Addis Ababa predominately expanded in these directions due to the expansion of transport infrastructure and subsequent urban expansion, as mentioned earlier. Clearly, the high percentage of urban expansion in these directions may be attributed to the size of quadrant, suitability of land and increase in the road network, which facilitated mobility and urban expansion, as indicated above. On the other hand, the analysis revealed that urbanization in the NNE, NEE and NNW directions was low compared to other directions. The low level of urban expansion in the direction of NNE, NEE, and NNW was attributed to the presence of physical barriers which is analogous with [53]. Maimaitiming and Zhang [53] stated that urban growth is mostly shaped by natural environments including topography and rivers. Similarly, Xu and Gao [22] contended that accessibility such as proximity to transportation, the CBD and amenities (e.g., distance to schools) is a universally important factor for urban expansion, while physical variables (e.g., terrain) significantly affect urban development in specific sectors due to the physical characteristics of the area. Indeed, physical characteristics such as topography limited the urban growth in the NNE, NEE and NNW directions in Addis Ababa.

In addition, the LEI analysis showed that the city demonstrated remarkably an outlying growth compared to others, implying that the urban patches created during the

study period were mainly a non-contagious. A more pronounced outlying growth was observed in the city over the past three decades correlates with similar studies conducted in other African cities such as Accra and Ghana [10] and contravene with the studies carried out in Beijing, Shanghai, Tianjin and Guangzhou [47]. Akubia and Bruns [10], in their study of spatio-temporal dynamics of land-use change and urban expansion in greater Accra Metropolitan revealed that the urban growth in Accra was characterized by uneven spatial development and differentiated an outward expansion. Similarly, Onuoha and Hu [54] analyzed urban growth patterns in Edwardsville/Glen Carbon, Illinois, using remote sensing population change data and LEI and concluded that the city predominately experienced an outlying growth between 1990 to 2015, reporting that outlying growth is not only observed in developing countries but also developed countries. On the contrary, Zhou and Wang [42] identified that Beijing, Shanghai, Tianjin and Guangzhou, demonstrated infilling and edge expansion, from 1990 to 2010, showing a spatial growth pattern that tended to be compact (coalescence). The highest proportion of an outlying growth implies that peri-urbanization is in progress in the city [26]. Similarly, Terfa and Chen [26], reported that the city of Addis Ababa showed a drastic increase in urban fragmentation, spatial irregularity and leapfrogging types of development due to the unplanned allocation of land for the city's key development projects such as the Integrated Housing Development Project (IHDP).The IHDP, which has been implemented since 2005 with an aim of addressing the housing demand of low and medium-income people significantly consumed the UGS in the city [55] and resulted in fragmented urban patches, which further contributed to an outlying urban growth pattern. This, compounded with the recent industrial park development, public and private investament and infrastructure expansion along eastern and southern arteries or edges of the city, far from the city center in the southern arteries, contributed to a predominant an outlying growth pattern. Evidently, the urban planning practices that aimed to facilitate the implementation of the IHDP and other development projects failed to ensure a compact city and rather resulted in dispersed urban growth which considerably reduced the UGS composition [4,26].

The rapid urban expansion in Addis Ababa in general, and in the outer zone in particular, was attributed to several factors, such as the change of government and introduction of free-market-oriented economics, implementation of Urban Development Policy (UDP), increasing housing rent price in the inner city areas, population increase, insecure tenure system, inner-city redevelopment program induced relocation of low income residents, weak control over public land increasing speculation and land prices [4,33,56]. These factors influenced the spatial growth of the city considerably over the study period. Specifically, the UDP, which stipulates the promotion of green development, expansion infrastructure, housing development, and establishment of Micro and Small-Scale (MSE) enterprise which creates job opportunities in urban areas and accelerated the demand for urban land subsequently led to rapid urban expansion. Similarly, improved transportation networks and infrastructural expansion in the southern, eastern, and southern parts of the city contributed to speedy urban expansion. Nonetheless, the rapid urbanization in Addis Ababa is not only attributed to the above factors but also the failure of urban land governance which stemmed the instability observed due to political transition observed in the country in the past three years. A recent study on the land and housing inventory carried out by the city administration revealed that an estimated 1338 ha has been illicitly occupied (might be greater than the planned expansion) by individuals, religious institutions and real estate developers [56]. Gloomily, an authorized land occupation has been observed in 88 (75%) districts (mainly in the outer districts), signifying unauthorized land occupation is seemingly the most complex urban land management challenge in Addis Ababa [56] and the failure of the city to address this challenge has contributed to the unprecedented level of decline of the UGS in recent years. This, combined with weak urban spatial plan implementation, frequent restructuring of public offices, poor enforcement of environmental laws, lack of transparency and accountability, and soaring land price and

informal land transactions, has made urban land governance in general and preserving UGS resources in particular, beyond the capacity of the city administration.

Finally, while this study improves our understanding of how urban areas have been expanded and UGS declined over the study period, it may not exactly show the actual pace and extent of change. This is because the study utilized a 30 m resolution image and cannot capture all the available UGS patches in the city. Therefore, future studies need to employ a Very High Resolution (VHR) to further investigate urban expansion and UGS change.

## 5. Conclusions

Since the end of the 1980s, the government has been implemented various policies to achieve orderly and planned urban growth in the country. The establishment of the National Urban Planning Institute in 1987, post-1991 decentralization of urban governance and spatial planning and the subsequent implementation of the UDP are some of major policy approaches that have been came into effect over the past decades. The aim was to achieve is to create green urban areas, where residents live, work and enjoy [33]. Some achievements have been reported over the past couple of years. Undeniably, housing stock have been increased, the quantity of the UGS increased in the inner zone, medium and small scale industries expanded, impressive economic progress have also been recorded in the city in the past few years, however, it has been at the cost of environment, because the UGS planning and management has been largely overlooked in development plans. Arguably, the result of this study has shown that the UGS in the city shrunk over the past decades, due to uncontrolled urban expansion. The lost 44% of its UGS during the study period to make a way for development. However, the change in the proportion of the UGS was non-unidirectional, i.e., increased and decreased in the study period, demonstrating a spatial trade-off between the two urban land use functions in the inner zone.

Over all, while there has been a growing number of studies on the spatiotemporal dynamics of urban expansion and UGS change, there has been a little effort on systematic assessment of the variation on the magnitude and directions of urban growth and UGS change in the city. The method applied in this study offered scientific evidence on spatial and temporal variation of urban growth and UGS change in the city from 1989 to 2019.The result showed that the city exhibited varying trend of change in the two zones and eights quadrants. Most importantly, the study also demonstrated that compared with the traditional spatial-temporal dynamic inquiry, zonal and directional along with LEI can give a better insight into urbanization dynamics and the pattern of UGS change, which is evidential for urban planners and decision makers for better urban planning.

In conclusion, while the city expanded outward over the study period, the UGS is generally declined in the city because the UGS have been sacrificed for economic development. A consistent decline of the UGS will result in environmental problems such as UHI, rises in temperature, pluvial flooding and reduce its ability to become climate resilient. Hence, a set of urban planning measures need to be implemented. First, future planning interventions need to adopt green space planning strategies (such as micro-level UGS planning) because Addis Ababa is running out of space, nowhere to expand or reclassify its boundary. Secondly, while preparing spatial plan, the existing standard developed by the national government should be strictly be adopted. Especially, the national urban green infrastructure standard which stipulates 30%, 30%, and 40% for UGS, road, and built up, respectively, need to be enforced. Thirdly, urban land management in Addis Ababa is apparently the most difficult task for the city government due to several social, economic, and political factors. Therefore, a new paradigm shift is required towards sustainable urban land management through wider stakeholder, community, and other interest group participation. Fourthly, as housing is becoming a chronic problem in Addis Ababa, the government should address the mounting housing problem and rising housing rental prices, which forced a significant number of city residents to seek shelter in the outskirt areas through illicitly occupying public land, and the UGS resources for environmental functional which aggravated the UGS decline.

**Author Contributions:** Conceptualization, Methodology, Software, Validation, Formal Analysis, Investigation, Data Curation, Original Draft Preparation, Writing E.M.W.–Supervision, Review and Editing P.V.G. All authors have read and agreed to the published version of the manuscript.

**Funding:** This research received no external funding.

**Data Availability Statement:** Data will be available up on request. Images employed for the study will be available online for readers.

**Acknowledgments:** I would like to acknowledge Addis Ababa City Planning Commission for providing secondary data.

**Conflicts of Interest:** The authors declare no conflict of interest.

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
