# Peer review of "Monitoring Urban Expansion and Urban Green Spaces Change in Addis Ababa: Directional and Zonal Analysis Integrated with Landscape Expansion Index"

_forests, doi:10.3390/f12040389_

Round 1

Reviewer 1 Report

Title

The title well reflects the substantive content of the article.

Aim

The purpose of the article is clear.

Methods

The adopted methodology of analyzes is a combination of several previously used and published methods. These methods seem to be well suited to the purpose of the research. The only issue for discussion is the choice of the concept of “directional analysis”. The division into eight direction quadrant angles seems to be “artificial”. However, it cannot be denied that the results of such analysis can be smartly presented.

Results and discussion

The results are clearly presented with reference to the subsequent stages of the analysis.

In the discussion section, I would suggest rethinking the statement (lines 429-430) which is a “…different trend of spatial growth compared to other countries in the same periods”. With reference to the articles cited, “other countries” tuned out to be only China.

Conclusions

They are not conclusions but a sort of a summary with an emphasis on the importance of the research carried out. Personally, I prefer synthetic conclusions that directly refer to the obtained results and, if necessary, the methods used. This allows them to be remembered more easily, which may be important for possible citations of the article. However, I leave these issues to the authors' decision.

Other comments

The text, including the list of references, requires careful editing (see, e.g., the lines 9, 119, 123, 209, 437-440)

Author Response

Dear reviewer

Dear Sir,

We thank you for the comments. I appreciate the time and effort that you have made in providing valuable comments on our manuscript.  We agree with all the comments and tried to make changes to the manuscript. We have also attached a separate file containing point by point response.

Eyasu Markso Woldesemayat

Reviewer 2 Report

The manuscript presents the study in which authors show how the urbanized area of Addis Ababa expanded while area of UGS shrinked over time. 

The innovative aspect of the study seem not to be enough emphasized. Authors compared the data from different years from the same, commonly available source. It seems that there should be more about what new was developed through this study, which could be utilized in other contexts around the world. 

Some more detailed comments:

How do authors define Urban Green Spaces? Line 48 suggests that what was considered in the study were peri-urban or even rural spaces which were absorbed by the expanding city fabric.

Figure 2 caption - can number of quadrants be 8, or just 4?

Some English grammatical and punctuation error scattered accoss the MS. English editing is suggested. 

Author Response

Dear reviewer

Dear Sir,

We thank you for the comments. I appreciate the time and effort that you have made in providing valuable comments on our manuscript.  We agree with all the comments and tried to make changes to the manuscript. We have also attached a separate file containing point by point response.

Eyasu Markos Woldesemayat

Reviewer 3 Report

I consider the article to be very interesting, orderly and logically thought-out. My review, concerns mainly methodological aspects, and these point to the authors' sound scientific workshop. 

The abstract captures the most important research results. 

In the keywords, I suggest a change: Addis Ababa, or urban green space into capital city (this term provides a broader context), green infrastructure and the addition of suburbanisation and spatial transformation. 

The introduction contains a well-drawn-out background for the urban transformation, the authors have defined what urban green space is. Line 99 - should be "this study". I think that in this part of the study, the authors should explain better why the analysis has been carried out over the last three decades? Why was 1989 taken into account when the capital was created in 1986? 

Chapter two „Methodolog” -  the description of the 'study area' as well as the 'sources of data' are well developed, supported by clear graphics. However, I wonder about the values quoted in lines 122-125. For example, line 125, does the increase in the built-up area from 24,942 ha to 35,050 ha really mean a 50% increase? In Table 2, the authors in some sense define different types of land cover. In my opinion, it is worth supporting these definitions, above all in relation to urban green and urban forest with specific definitions. It is not clear what 'thicket forest' or 'sparse vegetation' means. How have these spaces been identified/qualified with the adopted research tools. Urban green area also includes open sports grounds - sports fields, open swimming pools and so on. What group are these facilities qualified for. The presented way of dividing the land cover is therefore questionable and this element needs to be improved.  Chapter 2.3 presents in detail the formulas used to assess "urban growth and urban green change". I have the irresistible impression that by far the greatest effort was made by the authors to determine urban change, including those on a landscape scale. However, the green areas themselves, including urban forests, have not been given due attention, and there are only a few lines (234-242) that the greeness index was used for this purpose.

The „results" are well presented, in logical order in relation to the assumptions made in the methodology. The results have been included in tables and illustrated with interesting graphics. The „discussion” lacks a slightly broader context. There is a good explanation for the observations made. However, I would expect the discussion to provide some indication of the further directions of research and to inspire other researchers. Perhaps the addition of a limitation to the work will help to determine the future direction of research into the process of urbanisation of the area. I would also like to point out that the structure of the work is in a way similar to work Janeczko E. et al. (Sustainability 2019, 11, 3007; doi:10.3390/su11113007). That work also concerned three time intervals and changes within the metropolitan area, in addition, in the vicinity of the capital. So I think that it can therefore be helpful in discussing the results.

Author Response

(The authors gave the same response as above.)

Reviewer 4 Report

Please see comments in the attached file

Author Response

Dear reviewer

We have substantial changes in the text according to the comments. We thank you for your effort.

Eyasu Markos Woldesemayat

Round 2

Reviewer 2 Report

It looks like authors failed to address any of the previous comments successfully, regardless of extensive editing in their manuscript. 

Author Response

Date: 12/02/2021

To: Ms. Denise Li

Assistant Editor/MDPI

Ref: Manuscript ID: forests-1071723

Forest/MDPI

Dear Sir,

Thank you for giving me the opportunity to submit a revised version of our manuscript entitled “Monitoring Urban Expansion and Urban Green Spaces Change Addis Ababa: Directional and Zonal Analysis integrated with Landscape Expansion Index” to [Journal of forest/MDPI] for the second time. I appreciate your continuous effort to make the article meet your journal standard. We agree with all the comments and tried to make changes to the manuscript for the second time, as explained above. We are grateful to the reviewers for their insightfulness. We have also attached a separate file containing point by point response.

We look forward to seeing it published in your highly prestigious journal.

Kind regards,

Eyasu Markos Woldesemayat (*Corresponding Author)

Tianjin University, School of Architecture

Tianjin, China

Nankai District, Wei jin Lu road

Cell: +251912630022

E-mail: eyasumark@gmail.com